# The Role of Major Histocompatibility Complex in Organ Transplantation- Donor Specific Anti-Major Histocompatibility Complex Antibodies Analysis Goes to the Next Stage -

**DOI:** 10.3390/ijms20184544

**Published:** 2019-09-13

**Authors:** Tsukasa Nakamura, Takayuki Shirouzu, Katsuya Nakata, Norio Yoshimura, Hidetaka Ushigome

**Affiliations:** 1Department of Organ Transplantation and General Surgery, Kyoto Prefectural University of Medicine, Kyoto 602-8566, Japan; nyoshi@koto.kpu-m.ac.jp (N.Y.); ushi@koto.kpu-m.ac.jp (H.U.); 2Molecular Diagnositcs Division, Wakunaga Pharmaceutical Co., Led. 4-5-36 Miyahara, Yodogawa-ku, Osaka 532-0003, Japan; shirouzu_t@wakunaga.co.jp (T.S.); nakata_k@wakunaga.co.jp (K.N.)

**Keywords:** major histocompatibility complex, organ transplantation, donor specific anti-HLA antibodies, immunocomplex capture fluorescence analysis

## Abstract

Organ transplantation has progressed with the comprehension of the major histocompatibility complex (MHC). It is true that the outcome of organ transplantation largely relies on how well rejection is managed. It is no exaggeration to say that to be well acquainted with MHC is a shortcut to control rejection. In human beings, MHC is generally recognized as human leukocyte antigens (HLA). Under the current circumstances, the number of alleles is still increasing, but the function is not completely understood. Their roles in organ transplantation are of vital importance, because mismatches of HLA alleles possibly evoke both cellular and antibody-mediated rejection. Even though the control of cellular rejection has improved by recent advances of immunosuppressants, there is no doubt that antibody-mediated rejection (AMR), which is strongly correlated with donor-specific anti-HLA antibodies (DSA), brings a poor outcome. Thus, to diagnose and treat AMR correctly is a clear proposition. In this review, we would like to focus on the detection of intra-graft DSA as a recent trend. Overall, here we will review the current knowledge regarding MHC, especially with intra-graft DSA, and future perspectives: HLA epitope matching; eplet risk stratification; predicted indirectly recognizable HLA epitopes etc. in the context of organ transplantation.

## 1. Introduction

In order to protect living organisms, each of their component cells is equipped with a variety of barrier mechanisms. The internal environment of cells is closely monitored by the immune system via the major histocompatibility complex (MHC). Cell metabolism regularly decomposes old or unnecessary proteins, and fragments of these metabolites are presented on the cell surface. MHC captures these peptides and the immune system can subsequently recognize and analyze these molecules [1]. The role of MHC, therefore, is particularly important in organ transplantation, where non-self, normally allogeneic organs from one individual are transplanted into another individual. Antigen presentation by MHC can initiate various types of immunological rejection of transplants. Of these types of rejection, antibody-mediated rejection (AMR) has recently been paid much attention because it is clearly correlated with poor outcomes [2]. Both cellular and humoral immune systems are activated and initiated by CD4^+^ T cell allo-recognition via MHC (in human, MHC is named human leukocyte antigen (HLA)) [3]. On the other hand, generally, a major target of AMR is MHC. The antibodies involved are usually referred to as donor-specific anti-MHC (HLA) antibodies (DSA). The detection of DSA in serum (s-DSA) is generally required as a criterion for the diagnosis of AMR in heart, lung, liver, kidney, and intestinal transplantation. However, there are sometimes problems with s-DSA detection with regard to accuracy, sensitivity, specificity, and cost. Recently, alternative clinical methods have been applied to compensate for the disadvantages of s-DSA as outlined above, involving the detection of intra-graft DSA (g-DSA). In the present review of MHC, we will discuss the role of MHC in the context of organ transplantation with particular attention to AMR and g-DSA.

## 2. Major Histocompatibility Complexes (MHC)

### 2.1. History of Major Histocompatibility Complexes

The MHC is a family of genes that encode membrane proteins, which play important roles in the immune response. The MHC was discovered by Gorer and Snell et al. in 1936. Their skin transplantation experiments with mice revealed that self- and non-self recognition depended on the genetic background. Snell et al. named the group of mouse genes that determine self/non-self as histocompatibility-2 (H-2) [4,5]. Subsequently, it had been found that there was an antibody in the serum of a patient who received frequent blood transfusions [6], which caused other people’s white blood cells to aggregate. The antigen for this antibody is human MHC and was later named HLA [7]. Thus, Human MHC is referred to as HLA and mouse MHC is referred to as H-2.

There are multiple antigenic types of HLA antigens. Many of these serotypes, in particular HLA class I, have been analyzed based on the lymphocyte cytotoxicity test (LCT), which is devised by Terasaki [8]. Analysis of HLA class II began with the discovery of HLA-D by the mixed lymphocyte culture (MLC) test [9]. Subsequently, HLA-DR was identified as an antigen related with HLA-D by the LCT [10]. Later, another locus was found and referred to as HLA-DQ (previously DC). In addition, HLA-DP (previously SB) was identified by the primed lymphocyte test (PLT) [11]. At the present day, it is known facts that HLA-D is not a specific antigen but instead is composed of HLA-DR, DQ, and DP [12]. Several methods have been available for HLA typing, such as sequence specific primers (SSP), reverse sequence specific oligonucleotide probes (rSSOP) sequencing-based typing (SBT), and next-generation sequencing (NGS) [13,14].

### 2.2. Types of Major Histocompatibility Complexes

As described above, the HLA complex is encoded by several loci such as HLA-A, HLA-B, HLA-C, HLA-DR, HLA-DQ, and HLA-DP. These loci have many polymorphisms, so the combination (haplotype) is exceedingly large. However, the MHC exhibits strong linkage disequilibrium, which is the appearance of non-random association of alleles at multiple loci [15,16,17]. This linkage disequilibrium in the MHC region often causes a specific combination for each locus of MHC. When two genetic polymorphisms are present on the same chromosome, the two polymorphisms are classified as linked. Given that genetic recombination has occurred in a biologically conventional manner, polymorphisms at separate sites are not able to be determined as in the linked state. However, linkage disequilibrium is a state where certain gene polymorphism can be predicted with extremely high probability based on information of the polymorphism at a distant site. In the MHC, the gene loci are concentrated in a narrow region of chromosome 6 [18], so recombination between each gene is less likely to occur. Therefore, genes such as HLA-A, HLA-B, HLA-C, and HLA-DRB1 are often inherited in a linkage disequilibrium state. As HLA gene polymorphism analysis progresses, haplotypes that are associated with specific diseases [19,20,21] and that are frequently found in specific ethnic groups have been elucidated. These ethnic group-specific haplotypes are thought to be involved in the process of forming ethnic groups. Thus, these haplotypes are commonly used to search for ethnic roots [22,23,24]. MHC antigens can be broadly divided into two classes, class I and class II. They are classified based on the source of the presented antigen.

#### 2.2.1. MHC Class I

The MHC class I molecules are expressed on almost all nucleated cells and platelets, and their main function is to present endogenous antigens [14]. The MHC class I molecules are further classified into classical and non-classical groups. The classical class I molecules present antigen to T cells and the non-classical class I molecules have various functions and exhibit limited polymorphism. Classical MHC class I proteins have been subdivided as HLA-A, HLA-B, and HLA-C. On the other hand, HLA-E, HLA-F, HLA-G, MHC class I polypeptide-related sequence A (MICA) and FcRn etc. are classified as non-classical MHC class I [25,26,27].

#### 2.2.2. MHC Class II

The MHC class II molecules are expressed on antigen-presenting cells (APC) such as B cells, dendritic cells and macrophages, and their main function is to present exogenous antigens [20]. The MHC class II proteins have been identified as HLA-DR, HLA-DP and HLA-DQ. The MHC class II genes include HLA-DRA1, HLA-DQA1, HLA-DPA1 encoding α chain, HLA-DRB1, HLA-DRB3, HLA-DRB4, HLA-DRB5 (HLA-DRB3/4/5), HLA-DQB1, and HLA-DPB1 encoding β chain. HLA-DRA1 forms a heterodimer with HLA-DRB1 or HLA-DRB3/4/5. Similarly, HLA-DQA1 and HLA-DPA1 are also associated with HLA-DQB1 and HLA-DPB1, respectively [28,29]. The HLA-DR is divided into 5 groups consisting of DR1, DR51, DR52, DR53 and DR8 depending on the antigen group. The DR1 and DR8 groups both consist only of DRB1 as an expressed gene. On the other hand, The DR51, DR52, and DR53 groups contain DRB1 in common and furthermore consist of DRB5, DRB3, and DRB4, which is considered to be generated from *DRB1* gene by gene duplication, as expressed genes, respectively [30,31,32]. (Figure 1).

### 2.3. Structure of Major Histocompatibility Complexes

Comparing with the frequency of gene polymorphisms in other human genes, HLA gene polymorphisms are more frequent [33]. There are two types of gene polymorphisms, those with and without amino acid substitution. Proteins encode by gene polymorphisms with amino acid substitutions and those functions are drastically differently from original proteins, and may have adverse effects on human life. Therefore, gene polymorphisms with amino acid substitutions are rarely maintained. However, the ratio of polymorphisms with and without amino acid substitutions is nearly equal in the MHC region. It has been considered that it may be evolutionarily advantageous to have polymorphisms that alter the amino acids comprising the peptide binding cleft to obtain diversity against wide variety of antigens [34].

#### 2.3.1. MHC Class I

The MHC class I molecule is composed of a heavy chain containing three domains (α1, α2, and α3) and the β2 microglobulin (β2m) protein, which contains an immunoglobulin-like domain. The MHC class I molecule binds to an intracellularly digested peptide via its peptide-binding cleft composed of α1 and α2 domains (Figure 2A) [35]. The MHC class I molecule is unstable if the peptide is not bound, and takes on a stable structure upon peptide binding [36]. The gene encoding MHC class I is located on the short arm of chromosome 6, and gene polymorphisms, are concentrated particularly on exons 2 and 3 that encode parts of the α1 and α2 domains, which play a role in peptide binding [37,38,39,40,41]. The α chain of the classical MHC class I molecule has a transmembrane domain that facilitates its association with the cell membrane. The molecular weight of α chain and β2m protein is approximately 45 KDa, and 12 KDa, respectively. The α chain and β2m are expressed on the membrane surface in a non-covalently bound state [42].

#### 2.3.2. MHC Class II

The MHC class II molecule is composed of two domains, an α chain (α1, α2) and a β chain (β1, β2) (Figure 2B). It is bound to a peptide via its peptide-binding cleft, which is composed of α1 and β1 domains. Similar to class I MHC molecules, MHC class II molecules adopt a stable structure upon peptide binding. The nucleotide sequence encoding the MHC class II molecule is located on the short arm of chromosome 6, and the genetic information is encoded at a position closer to the centromere compared to the MHC class I genes [18]. The MHC class II molecule has transmembrane domains in both the α and β chains, and both are anchored on membrane at C-terminal region. The molecular weight of α chain and β chain is approximately 33 to 35, 27 to 29 KDa, respectively [43]. The α and β chains are non-covalently associated and expressed on the membrane surface [44,45].

### 2.4. Function of Major Histocompatibility Complexes

#### 2.4.1. MHC Class I

The MHC class I molecules present peptides derived from endogenous protein antigens formed by proteolysis of non-self cells such as virus-infected cells or tumor cells. These processed antigens are presented by binding to the peptide-binding cleft of MHC class I molecules. MHC class I molecules bind peptides approximately 8 to 11 amino acids in length [46]. Normally, T cells are not activated by peptides derived from self-cells, but they are activated by peptides from non-self cells that are displayed by the MHC class I molecules through T cell receptor (TCR). Thereafter, these activated T cells attack non-self cells and prevent the growth of non-autologous cells. NK cells harbor the killer inhibitory receptor that recognizes MHC class I and suppresses cytotoxicity [47]; therefore, NK cells only attack cells that do not express MHC class I.

#### 2.4.2. MHC Class II

The MHC class II molecules present peptide derived from an exogenous antigen. This peptide is generated by lysosomal degradation of exogenous foreign target cells such as bacteria or fungi that have been taken into the cell by endocytosis [48]. Peptides that bind to peptide-binding cleft of MHC class II molecules are longer than those that bind MHC class I molecules and are approximately 15 to 30 amino acids in length [49]. Once MHC class II molecules present antigen on the peptide-binding cleft and are recognized by CD4^+^ T cells through the TCR, the CD4^+^ T cells are activated and release various cytokines that stimulate Th1 cells and Th2 cells [50]. Then Th1 cells produce IL-2 and IFN-γ, which activate CD8^+^ cytotoxic T cells and NK cells [51]. Cytotoxic T cells release cytotoxic granules containing perforin or granzymes to the foreign target cells. Perforin polymerizes on the surface of the foreign target cells to create pores, and granzymes induce apoptosis in the target cells [52]. Th2 cells produce cytokines such as IL-4, which activates B cells. Activated B cells produce antibodies against target cells and opsonize these cells [53]. Opsonization is either mediated by the complement system or by an antibody. Complement opsonization occurs when there is a large amount of IgM antibody [54]. Complement binds to the IgM antibody, which is bound to the foreign target cells, and C5a is produced. C5a then activates phagocytes to ingest the target cells. Antibody opsonization occurs when there is a large amount of IgG antibody. The IgG antibodies bind to the foreign target cells, and recruit phagocytes via Fcγ receptor that is bound to the IgG antibodies. The foreign target cells are then ingested by the phagocytes [55].

## 3. Major Histocompatibility Complex in Organ Transplantation

The MHC plays a dominant role in transplant rejection. On the other hand, the roles of minor histocompatibility antigens are also crucial and are reviewed elsewhere [56]. Following reperfusion of an allograft, allogeneic recognition is inevitably initiated. Ischemia-reperfusion injury can potentially initiate and accelerate subsequent rejection via damage to tissues [57]. Here, importantly, both donor-derived and recipient-derived APC may be the instigator of rejection. There are three potential pathways. In a unique aspect of transplantation, donor-derived APC directly adduce donor-derived antigens by MHC to recipient-derived CD4^+^ T cells; this is known as the direct pathway [58]. Recipient APC process donor cells and present certain molecules via their MHCs to recipient-derived CD4^+^ T cells; this is referred to as the indirect pathway [59]. Interestingly, non-processed donor MHCs can be utilized on the surface of recipient APCs and recognized by recipient-derived CD4^+^ T cells; this is known as the semi-direct pathway [60] (Figure 3). These various pathways involving MHC orchestrate allograft rejection.

### 3.1. Major Histocompatibility Complexes in Cellular Rejection

In cellular rejection, recently it has been reported that many types of immune cells are involved, such as neutrophils [61], natural killer cells [62], or myeloid derived suppressor cells [63,64] etc., besides T lymphocytes. Nevertheless, still T lymphocytes play the central role. The cytotoxic effects of CD4^+^ T cells seem to be not basic features of immune systems, although CD4^+^ T cells are of importance in initiating allograft rejection [65]. Following CD4^+^ T cell recognition, CD8^+^ T cells are generally activated through the binding of MHC class I and CD8 TCR. Alloreactive cytotoxic CD8^+^ T cells are subsequently generated and then reject allografts. The three distinctive pathways, direct, indirect, and semi-direct, begin the recognition of alloantigens by MHC class II and CD4 TCR. Generally, it has been reported that direct pathway strongly induces alloantigen-specific cytotoxic CD8^+^ T cells [66]. However, the directly, indirectly, and semi-directly activated T cells communicate with each other. The following three- and four-cell cluster models have been proposed. In circumstances where donor parenchymal cells react directly with CD8^+^ T cells, recipient CD4^+^ T cells activated by the indirect pathway help to effectively activate these CD8^+^ T cells, provided that secondary lymphoid tissue exists [67,68]. In the four-cell cluster model, CD4^+^ T cells activated both directly and indirectly cooperate and suppress towards rejection, especially indirect pathway tends to lean to immunoregulation side [69]. On the other hand, recipient CD4^+^ T cells activated indirectly cooperate in the semi-direct activation of CD8^+^ T cells. In addition to this, CD4^+^ T cells activated by the same APC which have acquired both donor-MHC and self-MHC class II cooperate with each other [68]. According to these models, each pathway can intricately communicate and induce allo-responses. Under the strict surveillance of the immune system, just a single MHC class II mismatch has the potential to cause cellular rejection, although the intensity of any rejection may depend on the organs or tissues involved. For example, skin transplant rejection is generally more severe than cardiac allograft rejection, probably due to the content of APC [70].

In order to reduce MHC-induced immunological rejection, transplantation models have been tried where MHC class II expression was knocked down using small interfering RNAs (siRNAs). These models have shown that even though alloreactive CD8^+^ T cells attack MHC class I molecules on grafts, it is clear that CD4^+^ T cell activation by MHC class II is required [71]. The modulation of MHC expression by siRNAs during ex vivo perfusion might be an effective method for attenuating cellular rejection during the acute phase [72].

Another cellular rejection pathway involving CD4^+^ T cells is the delayed-type hypersensitivity (DTH) reaction. Alloreactive CD4^+^ T cells activate macrophages via IFN-γ, resulting in DTH-mediated rejection. It has been reported that this type of rejection plays a central role in corneal transplantation [73].

There is an additional type of cellular rejection: plasma cell-rich rejection. Although plasma cells are differentiated B cells, which are responsible for antibody-mediated rejection, plasma cell-rich rejection is categorized as cellular rejection. However, there are several important differences between plasma cell-rich rejection and typical T cell-mediated cellular rejection [74]. Importantly, plasma cell-rich rejection is typically refractory to rejection therapy and associated with poor graft survival [75]. Although it shares a histological similarity with T cell-mediated rejection, plasma cell-rich rejection is associated with Th2 polarization and the presence of DSA [75,76]. Thus, plasma cell-rich rejection is categorized as a distinctive entity. Any association between AMR and plasma cell-rich rejection would be an interesting topic to explore from the perspective of treatment and pathogenesis.

### 3.2. Major Histocompatibility Complexes in Antibody-Mediated Rejection

Generally, through the indirect pathway, B cells present processed alloantigens to CD4^+^ T cells in a self-MHC-restricted manner. Then, humoral allo-immunity is initiated by indirectly primed CD4^+^ T cell and B cell activation and generating long-lived plasma cells [77,78]. Although a variety of alloreactive antibodies are generated, DSA have attracted the most attention because they are in the majority and have significant outcomes.

AMR can be divided into two types, depending on the time course of onset: acute and chronic. In the field of kidney and liver transplantation, acute and chronic AMR can be clearly distinguished based on pathological findings [79,80]. There is no clearly articulated differential definition in other fields, such as heart, lung, pancreas, and small bowel transplantation, although diagnostic criteria for AMR do exist [81,82]. Generally, class I DSA seem to play an important role in crossmatch positive transplantation, while de novo class II DSA have a significant impact on graft loss in the long-term following transplantation [83]. Theoretically, class II antigens rarely exist in most graft tissues, excluding endothelial cells, smooth muscle cells, and APC, whereas class I antigens are widely expressed. Therefore, there are abundant targets for class I DSA, but those for class II DSA do not exist plentifully, so any subsequent AMR is mainly evoked through the presence of class I DSA. However, it is also true that there is variation in the expression of class II antigens in pre-transplant tissues, which may depend on circulating cells. The initial expression of class II HLA antigens might determine transplantation outcomes [84]. On the other hand, following persistent tissue injury arising from rejection, infection, or drug toxicity the up-regulation of class II antigens might occur and they then become a target for class II DSA [85,86,87,88]. AMR mediated by class II DSA seems to present a persistent and considerable burden on graft function in kidney [89], liver [90], heart [83], lung [91], compared with rejection due to class I DSA.

In clinical transplantation, although minimization of calcineurin inhibitors is preferred because of side-effects, below acceptable levels of immunosuppression finally leads to chronic AMR and results in graft-loss [92]. Thus, from this point of view, it is reasonable to assume that the control of de novo DSA, especially class II DSA, is a key factor if positive long-term outcomes in organ transplantation are to be achieved.

## 4. Analysis Methods of Anti-Major Histocompatibility Complex Antibodies

In this section, we will review methods which are frequently applied in the field of organ transplantation. These techniques are also well reviewed in elsewhere [93]. To detect humoral factor: DSA, originally, LCT or the complement dependent cytotoxicity (CDC) crossmatch test was reported in the 1960s (8). LCT was widely applied from then. However, relatively low sensitivity of LCT sometimes did not elicit the existence of DSA. Then, the flow cytometry crossmatch test (FCXM) or methods to detect DSA by using refined HLA antigens were introduced.

### 4.1. Lymphocyte Cytotoxicity Test (LCT)

LCT detects an antigen-antibody and complements dependent cytotoxicity reactions. Donor lymphocytes and the recipient serum are incubated together, followed by the addition of rabbit complements. Subsequently, lymphocytes are necrotized by the complement-dependent cytotoxicity reaction, provided that DSA exist. By adding eosin dye, necrotized lymphocytes are observed by a phase-contrast microscopy. In order to improve sensitivity, there is an alternative method that anti-human globulin (AHG) is added after a mixed reaction, followed by the addition of rabbit complements: AHG-LCT.

### 4.2. Flow Cytometry Crossmatch (FCXM)

Following the reaction between recipient serum and donor lymphocytes, PE-conjugated anti human IgG is added as a secondary antibody. Then, this reaction is analyzed as a fluorescent intensity by flow cytometer. FCXM can also detect non-HLA antibodies, but cannot estimate the degree of cell toxicity.

### 4.3. Immunocomplex Capture Fluorescence Analysis (ICFA)

Immunocomplex capture fluorescence analysis (ICFA) is one of the crossmatch techniques by utilizing WAKFlow HLA antibody class I and II ICFA (Wakunaga Pharmaceutical Co., Ltd, Osaka, Japan). DSA immunocomplex can be detected by this technique with high specificity. Luminex xMAP technology (Luminex Corporation, Austin, TX) is applied as a detection system. First, HLA and DSA immunocomplexes are formed after the reaction between recipient serum containing DSA and donor lymphocytes. Subsequently, lymphocytes that have formed immunocomplexes are lysed and the immunocomplexes and remain in lysates. Next, anti-HLA monoclonal antibodies immobilized on Luminex beads capture these immunocomplexes. Finally, Luminex system detects these immunocomplex/Luminex beads bounded by PE-conjugated anti-human IgG.

### 4.4. Human Leukocyte Antigens (HLA) Antibody Testing

Refined HLA antigens from human lymphocytes are immobilized on micro-beads. These micro-beads are incubated with recipient serum. Then, the reaction is detected by flow cytometer or Luminex system. According to types of refined HLA antigens, there are different methods: panel reactive antibody (PRA); single antigen beads assay etc.

#### 4.4.1. FlowPRA

The FlowPRA screening test is performed to detect anti-HLA antibodies in the serum. FlowPRA uses latex beads coated with mixed class I or class II HLA antigens purified from panel cells. About 30 types of panel cells are selected for class I and class II, respectively, to cover typical types of HLA antigens (One Lambda Inc, Canoga Park, CA). First, a reaction is caused between anti-HLA antibodies and these beads. Next, FITC-conjugated anti-human IgG is added. Then, mean fluorescence intensity (MFI) is measured by flow cytometry and the shift from the negative control is calculated. The specificity of anti-HLA antibodies cannot be identified by FlowPRA, because each bead contains multiple HLA antigens. 

#### 4.4.2. Single Antigen Beads Assay (SAB)

Single antigen beads assay (SAB) is performed to determine the specificity of the anti-HLA antibody. SAB kits consist of Luminex beads coated with a single HLA antigen extracted from gene-modified cells. First, a reaction is performed between the anti-HLA antibody and SAB beads. Next, PE-conjugated anti human IgG is added as a secondary antibody. Following these reactions, signal levels of MFI of PE were measured by Luminex system (Luminex Corporation). SAB kits are commercially available from One Lambda Inc, Immucor Inc (Norcross, GA) or Wakunaga Pharmaceutical Co., Ltd.

### 4.5. Complement Fixation Test

Activation of the complement system following an antigen–antibody reaction is a necessary step in AMR-mediated graft injury. Thus, the estimation of complement activation can act as an AMR progression marker. In the field of transplantation, C4d, C1q, and C3d, which are metabolites of complement, are often utilized as markers. First, the accumulation of C4d, which connects with target tissues via covalent bonds [94], is currently recognized as a sign of an allogeneic reaction mediated by antibodies. Immunohistochemical analysis is routinely used to detect C4d deposition in an allograft. Clinically, C4d detection is the current “gold standard” method [79,80,81,82]. Originally, Feucht et al. [95] reported that C4d was deposited in the peritubular capillaries of renal allografts in cases of hyper-acute AMR. The immunohistochemical detection of C4d provides evidence of an AMR reaction. Second, C1q, as an initiator of the complement activation pathway, is an important marker for estimating the degree of DSA-mediated tissue injury. The existence of C1q fixing DSA correlates with a higher risk of renal allograft loss [96]. On the other hand, although DSA has a negative impact, C1q status does not reflect a significant risk to patient survival following liver transplantation [97]. Third, C3d is also considered to be an important marker of complement activation. C3d, a terminal metabolite of immunocomplexes, could be a predictor of allograft loss due to AMR [98]. The binding ability of C1q and C3d is generally analyzed based on a solid-phase assay. These assays are considered to assess the degree of harm DSA have caused to an allograft.

## 5. Graft ICFA, Intra-Graft Donor Specific Anti-HLA Antibodies

Logically, the assessment of g-DSA seems to have an advantage over the assessment of s-DSA when considering AMR due to DSA. Recently, therefore, much attention has been paid to g-DSA. Research of g-DSA is summarized in Table 1. In the early era of organ transplantation, assessment of g-DSA was attempted using the methods available at the time. In 1972, Metzer et al. [99] attempted to examine eluates from rejected kidney allografts. They reported that there were significant differences in serum and intra-graft DSA activity. So, these differences in s- and g-DSA have long been recognized. In the early part of this century, chronic allograft nephropathy remained a leading cause of allograft loss in renal transplantation, despite the introduction of calcineurin inhibitors. Martin et al. [100] reported that both class I and II DSA were detected in eluates from allografts with chronic nephropathy, and that these DSA played an important role in the progression of this type of lesion. Next, a concern about the distribution of g-DSA in renal allografts was investigated by assessing g-DSA levels in the cortex and medulla [101]. As a result, the distribution of g-DSA was concordant in 7/10 cases, but there were a few discrepancies between these sites. Thus, it is reasonable to analyze both the cortex and medulla when g-DSA levels of renal grafts are assessed.

More recently still, Taupin and colleagues reported g-DSA detection by acid elution techniques in kidney [102], liver [103] and lung transplantation [104]. The results of these reports are almost concordant: the presence of g-DSA results in poor graft survival and correlation is intense rather than s-DSA. In the field of renal transplantation, the relationship between DSA and AMR has been clearly demonstrated. However, there is insufficient evidence to identify a similar relationship in the context of liver transplantation. Neau-Cransac et al. [103] reported a relationship between g-DSA and AMR in a patient who underwent two consecutive liver transplantations. In this case, several types of g-DSA against the same epitope seemed to accumulate following the first liver graft, while s-DSA fluctuated and showed no constant trend. Although there are diagnosis criteria for AMR in liver transplantation, it is sometimes difficult to correctly diagnose AMR in a clinical setting due to the existence of many modifiers. Thus, it is suggested that the detection of g-DSA may help to distinguish AMR from other lesions.

Similarly, in lung transplantation, the diagnosis of AMR can sometimes be challenging. Colleagues in Bordeaux tried to identify g-DSA following lung transplantation. They concluded that g-DSA are an effective biomarker to identify harmful DSA [104].

Furthermore, research into g-DSA has also included epitopic characterization [105]. This research demonstrated that g-DSA are directed against allografts at epitopic levels, but not necessarily at antigenic levels.

Thus far, dissociation techniques have seemed to be the only way to detect g-DSA. However, Nakamura and colleagues attempted to detect HLA-DSA complexes without using a dissociation process. This method is called graft ICFA, which captures HLA-DSA complexes using a sandwich technique. Specifically, ICFA in a serum crossmatch test was applied for g-DSA assessment. If reactive DSA exist and reach HLA in an allograft, an HLA-DSA complex is already formed in the recipient. Therefore, this method has clear advantages in its reasonableness and speed. However, because the ICFA technique recognizes a non-epitope region of the HLA, graft ICFA is currently only able to distinguish between subclasses of HLA: HLA class I, class II DR, DQ, and DP.

In 2017, the graft ICFA technique was introduced in the field of transplantation for the first time [106]. High levels of g-DSA accumulation were observed in patients with chronic active AMR. It has previously been suggested that DQ-DSA is one of the most important antibodies for treating chronic active AMR. In fact, in clinical settings s-DQ-DSA are often found in patients with chronic active AMR, although it is thought that HLA-DQ antigens are rarely expressed in the kidney. In this research, graft ICFA was also shown to be an effective tool for identifying AMR during the early phase, when s-DSA cannot be detected. Furthermore, g-DSA can be removed through the appropriate treatment of acute AMR. Thus, g-DSA could have the potential to be both a diagnostic criterion and a therapeutic gain factor.

In other research [107], the clinical course of DSA+ renal transplantation was analyzed with the results of graft ICFA. Following renal transplantation and de-sensitization, residual s-DSA reacted, manifested transient AMR, and were gradually eliminated from the allograft in a case of successful de-sensitization. Again, g-DSA could also be a useful marker for determining AMR in crossmatch positive transplantation.

Recently, g-DSA levels in patients with chronic active AMR were also analyzed following various treatments, including anti-CD20 antibodies, steroid pulses, and plasmapheresis. Nakamura et al. [108] found that patients with relatively lower g-DSA levels tended to exhibit therapeutic responses following g-DSA clearance, while there were no significant differences in pre-treatment Banff scores in the non-response group. Thus, g-DSA level might be a useful index for determining whether effective de-sensitization can be performed in patients with chronic active AMR.

In order to accelerate techniques for g-DSA analysis and control of AMR, basic research in rodent models is always important. Graft ICFA in a mouse cardiac transplantation model has been recently established, especially class I [109]. Cardiac transplantation of a sensitized individual was used and g-DSA levels were assessed, mainly in the acute phase of cardiac transplantation. Although sensitized models had both class I and II DSA, only class I DSA were detected as g-DSA. They pointed out that cardiac allografts expressed only class I DSA, and not class II, in the acute phase. Therefore, g-DSA comprised class I DSA only. From this perspective, it is reasonable to recognize class I DSA as hazardous DSA when considering cardiac transplantation indications. However, it should be kept in mind that APC are also included in the allograft and MHC expression would change over time following transplantation. Class II MHCs might be expressed following the accumulation of allograft tissue injuries. With the introduction of this method, it can be expected that basic medical research for AMR will progress through the use of mouse transplantation models.

Interestingly, they also reported that the graft ICFA technique can be used for the detection of A/B antigens and anti-A/anti-B antibodies, or ABO-DSA complexes in allografts [110]. This technique could be valuable in assessing AMR due to ABO-DSA, because anti-A/anti-B antibodies may not increase in the serum during the acute phase of AMR.

The graft ICFA technique could be expanded to detect non-HLA antibodies by adding beads against specific target antigens. It might become possible that a variety of g-DSA is detected simultaneously from tiny graft specimens. Graft ICFA technique seems to have a lot of potential and might be a standard method to investigate AMR. Further developments in g-DSA field are expected.

## 6. Future Perspective

In the current circumstances, it is true that clinical organ transplantation is performed according to the results of a series of crossmatch tests, as discussed in this review. Whether sensitized pair transplantation can be performed or not largely depends on DSA MFI following desensitization, although this is not essential. Ideally, bespoke risk stratification would be performed for each pair so that irreversible rejection may be avoided. In this section, up-to-date trends are discussed as a feature perspective.

### 6.1. HLA-Epitope Matching

In general, DSA recognize a part of the HLA three-dimensional structure in the size-range of 3 Å. As mentioned above, DSA tend to react with allografts in an epitope-restricted manner. Epitopes normally consist of three consecutive amino acids. In other words, an identical antibody can react with more than one HLA antigen when these antigens possess the same epitope. Antigens which possess the same epitope are referred to as cross-reacting groups (CREGs). Thus, in organ transplantation, much attention has been paid to this HLA-epitope matching as a topic of interest beyond serological MHC matching. HLAMatchmaker provides information about the reactivity of anti-HLA antibodies at the epitope level [124,125]. These epitopes should be considered prior to transplantation, so that DSA can be avoided.

### 6.2. HLA-Eplet Matching

Distantly located amino acids sometimes form an epitope, which is referred to as an eplet. Eplet analysis can now also be taken into consideration due to recent developments in the understanding of the stereochemical configurations of DSA and specific HLA antigens. An eplet version of HLAMatchmaker is available and provides HLA compatibility information [126]. In clinical transplantation, we often experience the situation where multiple types of anti-HLA antibodies, and not only against the donor HLA, are produced. This phenomenon can be explained by these eplets or epitope-based DSA production. The analysis of histocompatibility at the eplet level may lead to techniques which allow sufficient control of humoral allo-immunity.

### 6.3. Predicted Indirectly Recognizable HLA Epitopes

Although HLAMatchmaker can estimate potential DSA production, T cell reactivity against HLAs remains unknown. The reaction of anti-HLA antibodies to HLA molecules is generally predictable based on HLA alleles. This predictability is reliable in the case of class I HLA molecules, since anti-HLA antibodies usually recognize long peptides as their target-binding region, and with a specific anchor amino acid sequence. It is difficult to predict T cell reactivity against mismatched HLAs [127], although allopeptide recognition via the indirect pathway by T cells plays an important role in developing DSA and AMR [77]. A phenomenon exists where a peptide derived from a specific class I HLA tends to bind firmly to a cell expressing certain class II HLA molecules [128]. This implies that there is a relationship between peptides derived from donor HLAs and recipient class II HLAs, resulting in the modulation of T cell reactivity and subsequent DSA production. Predicted indirectly recognizable HLA epitopes (PIRCHE) are targets for T cells in the indirect allo-recognition pathway. The PIRCHE algorithm can predict donor HLA-derived peptides shown by HLA-DRB1 of recipient antigen presented cells [129]. Mismatched HLA epitopes presented on class I HLAs are defined as PIRCHE-I, while PIRCHE-II denotes HLA epitopes presented on class II HLA molecules. Generally, PIRCHE-I can be used to estimate CD8+ T-cell allo-reactivity, whereas PIRCHE-II can be used to estimate CD4+ T cell reactivity. It has been reported that the PIRCHE-II scoring system might be related to the incidence of de novo DSA production and subsequent allograft survival [130]. This PIRCHE approach could go beyond classical antigen matching.

### 6.4. Direct Crossmatch Test by Graft ICFA

As discussed in 6.1–6.3, our core concern is whether an allograft will be rejected by DSA. It is not enough to perform serum and lymphocyte crossmatch tests because antigens expressed on lymphocytes are not necessarily identical to those of allografts. It is also true that there is not enough time to perform crossmatch tests using tissues from prospective allografts and sera by the graft ICFA technique in cadaveric transplantation, and that they are too invasive for live-donor transplantation. However, this might be the most reliable method for predicting AMR. In the future, it may be possible to generate in advance cells from prospective allografts using, for example, induced pluripotent stem cells. In order to perform organ transplantation free from AMR, direct crossmatch tests by graft ICFA would be highly beneficial.

## 7. Concluding Remarks

With the development of analytical methods, many phenomena regarding HLA and DSA can be predicted and explained. Ideally, the risk of developing DSA can be avoided or reduced, provided a donor who shares the identical epitopes is selected. This selection method might be possible in cadaveric transplantation with development of the integrated epitope database. However, epitope matching is not realistic in many situations, including emergency or unrelated live-donor transplant etc. At the same time, it is true, therefore, that we should take measures to manage AMR appropriately.

## Figures and Tables

**Figure 1 ijms-20-04544-f001:**
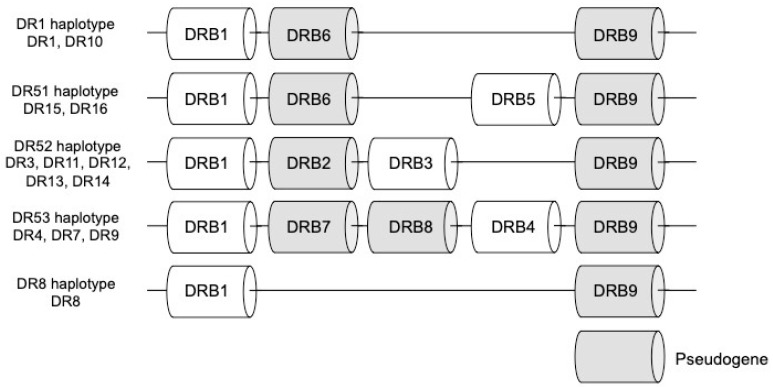
The human leukocyte antigen (HLA-DR) region is divided into five groups, DR51, DR52, DR53, DR1, and DR8, depending on the number and combination of genes. Gray box indicates pseudogene.

**Figure 2 ijms-20-04544-f002:**
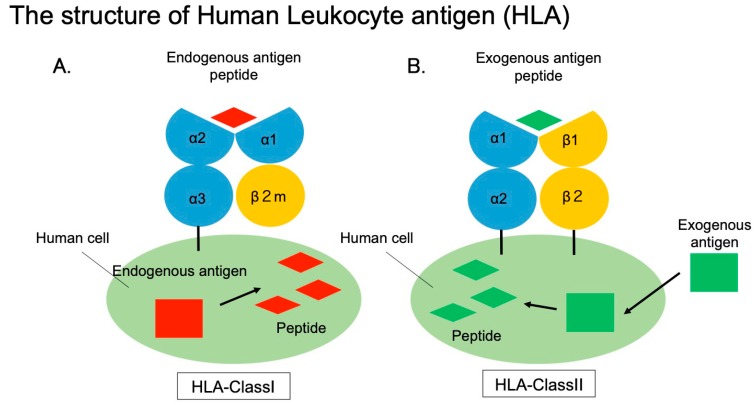
**A**. Structure of HLA class I molecules. Endogenous antigens such as tumor cells and infected cells are presented as peptide. **B**. HLA class II structure. Exogenous antigens taken up by phagocytic cells are presented as peptide.

**Figure 3 ijms-20-04544-f003:**
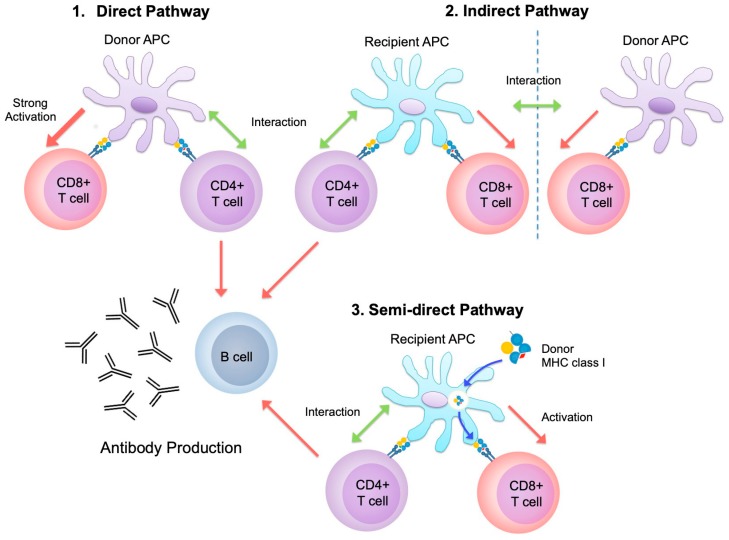
1.Direct pathway: CD4^+^ T cells activate and generate allograft-specific cytotoxic CD8^+^ T cells and humoral allo-immunity via donor antigen-presenting cells (APC). (Three-cell cluster). 2. Indirect pathway: recipient APC process donor antigens and present these molecules and activate CD4^+^ T cells, subsequently linked (Three-cell cluster) or non-linked CD8^+^ T cells (Four-cell cluster) can be activated by these CD4^+^ T cells help. 3. Semi-direct pathway: intact donor MHC class I molecules are reused by recipient APC.

**Table 1 ijms-20-04544-t001:** Assessment of intra-graft donor-specific anti-MHC (HLA) antibodies (DSA).

Author	Refs.	Year	Species	Organ	Sample	Methods	Detector	Remarkable Findings
Metzger	[99]	1972	Human	Kidney	removed grafts	Acid Elution	LCT/MLC	Well antibody activity could be recovered from allografts with hyper acute antibody-mediated rejection.
Pedersen	[111]	1974	Sheep	Kidney	removed grafts	Acid Elution	LCT/MLC	DSA were bound to graft antigens during the rejection process.
Jeannet	[112]	1975	Human	Kidney	removed grafts	Acid Elution	LCT/MLC	The balance between intra-graft cytotoxic and blocking factors might determine the outcome of allografts.
Moschi	[113]	1979	Dog	Lung	removed grafts	Acid Elution	LCT/MLC	A considerable amount of DSA was confirmed in recipients without immunosuppression.
McPhaul	[114]	1981	Human	Kidney	removed grafts	Acid Elution	LCT/MLC, IH	g-DSA contained two types: 1. cytotoxic Abs to mononuclear cells; 2. Abs with specificity for kidneys.
Mohanakumar	[115]	1981	Human	Kidney	removed grafts	Acid Elution	LCT/MLC, IH	Rejected allograft contained multispecific alloantibodies, not only reactive to MHC class I and II.
Joyce	[116]	1988	Human	Kidney	removed grafts	Acid Elution	LCT/MLC, IH	Eluted DSA recognized organ-specific antigens expressed on the kidney cells.
Lucchiari	[117]	2000	Human	Kidney	removed grafts	Acid Elution	LCT/MLC, FCM	Eluted antibodies activated human endothelial cells, resulting in upregulation of adhesion molecules.
Martin	[100]	2003	Human	Kidney	removed grafts	Acid Elution	FCM	The detection rate of intra-graft DSA is greater than in serum before the removal of chronic rejected allografts.
Zou*	[118]	2006	Human	Kidney	removed grafts	Acid Elution	Luminex	MICA-DSA were detected in allografts of patients on transplantation waiting list.
Heinemann	[119]	2006	Human	Kidney	removed grafts	Acid Elution	ELISA/Luminex	Allografts harbor DSA, including non-complement binding DSA.
Bocrie	[101]	2007	Human	Kidney	Biopsy	Acid Elution	Luminex	The distribution of intra-graft DSA between the cortex and medulla is roughly concordant.
Heinemann	[120]	2007	Human	Kidney	removed grafts	Acid Elution	ELISA/Luminex	Allografts harbor DSA, including non-complement binding DSA.
Martin	[121]	2010	Human	Kidney	Biopsy	Acid Elution	FCM	Graft eluates contained non-DSA. The rate of detecting s and g-DSA is almost the same in patients with graft dysfunction.
Bachelet	[102]	2013	Human	Kidney	Biopsy	Acid Elution	Luminex	g-DSA, not s-DSA, are a severity and prognostic marker of AMR.
Neau-Cransac	[103]	2015	Human	Liver	Biopsy	Acid Elution	Luminex	AMR detected as g-DSA deposition in liver allograft might explain graft dysfunction.
Milongo	[105]	2016	Human	Kidney	removed grafts	Acid Elution	Luminex	g-DSA are generally directed against the donor at an epitopic level.
Visentin	[104]	2016	Human	Lung	Biopsy	Acid Elution	Luminex	The presence of g-DSA means a higher risk for graft loss.
Nakamura	[106]	2017	Human	Kidney	Biopsy	ICFA	Luminex	Graft ICFA is a useful technique to make an early and accurate diagnosis of AMR.
Nakamura	[107]	2017	Human	Kidney	Biopsy	ICFA	Luminex	g-DSA measured by graft ICFA are a marker of effective de-sensitization in crossmatch positive renal transplantation.
Norcera	[122]	2017	Human	Kidney	Biopsy	Acid Elution	Luminex	The presence of g-DSA indicates clinically relevant antibodies which should be monitored.
Courant	[123]	2018	Human	Kidney	Biopsy	Acid Elution	Luminex	Results of this study did not associate g-DSA with graft loss.
Nakamura	[93]	2019	Human	Multiple	Biopsy, removed organs	ICFA	Luminex	g-DSA in heart, lung, liver, pancreas and intestine as well as kidney grafts are also detected by graft ICFA technique.
Nakamura	[108]	2019	Human	Kidney, Liver	Biopsy	ICFA	Luminex	g-DSA measured by graft ICFA are a marker to predict therapeutic responses in chronic active AMR recipients.
Nakamura	[109]	2019	Mice	Heart	Biopsy	ICFA	Luminex	Graft ICFA can be applied in mice transplantation models. In the acute phase, class I DSA play important roles.
Nakamura**	[110]	2019	Human	Kidney	Biopsy	ICFA	Luminex	ABO-DSA can also be detected by graft ICFA technique.

*detection of intra-graft anti MICA antibodies, ** detection of intra-graft anti A/B antibodies, Abbreviations: Abs: antibodies; DSA: donor-specific anti-MHC (HLA) antibodies; ELISA: enzyme-linked immunosorbent assay; FCM: flow cytometer; g-DSA: intra-graft DSA, HLA: human leukocyte antigens; IH: immunohistochemistry; LCT: lymphocyte cytotoxicity test; MHC: major histocompatibility complex; MICA: MHC class I polypeptide-related sequence A; MLC: Mixed lymphocytes culture.

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
