# Peer review of "The Role of Major Histocompatibility Complex in Organ Transplantation- Donor Specific Anti-Major Histocompatibility Complex Antibodies Analysis Goes to the Next Stage -"

_ijms, 2019, doi:10.3390/ijms20184544_

Round 1

Reviewer 1 Report

This review is basically well-written and gives an extensive overview on the role of MHC and HLA-directed immunity in clinical transplantation. However there are some issues that need to be addressed.

General issues

The Part on MHC is too extensive, it should be better focused on clinically relevant aspects.

In particular, there is too much room for the immunological basis, in relation to a critical consideration of the clinically relevant methods in the field.

Specific issues

Introduction, page 2, line 7: “evidence to suggest that g-DSA might play an important role in the diagnosis, prognosis,…” this statement appears somewhat exaggerated, since the method has been published exclusively by the review authors themself. In addition, the fact that the pharmaceutical company marketing this method is represented among the authors, is raising doubts on the objectivity of this statement.

Page 2, line 31 “Currently, the main HLA identification tool is the DNA typing technique (rSSOP method) based on the Luminex system (14)”.

The basis of this statment is not clear: The reference is aged 10 years. It is not clear whether the statement refers to high- or low-resolution typing. NGS in case of HR typing, and rSSOP as well as SSP low resolution should be mentioned, as well as the practicability and suitability of the respective methods with respect to the available time in the clinical setting.

page 3, lines 12, 23

“On the otherhand, HLA-E, HLA-F, HLA-G, NKG2D ligand receptor and FcRn are classified as nonclassical MHC class I (27-29).”

NKG2D does not belong to the non-classical MHC class I. Instead, it is a receptor for the latter.

Author Response

Dear Reviewer 1

Thank you for the valuable and helpful comments, which have helped us to improve the manuscript.

Point 1. This review is basically well-written and gives an extensive overview on the role of MHC and HLA-directed immunity in clinical transplantation. However there are some issues that need to be addressed.

General issues

The Part on MHC is too extensive, it should be better focused on clinically relevant aspects.

In particular, there is too much room for the immunological basis, in relation to a critical consideration of the clinically relevant methods in the field.

Response 1:

We appreciate your valuable comment. In order to reduce the volume, we deleted the sentences in 2.1.History of Major Histocompatibility Complexes, 2.2.1. MHC class I, 2.2.2. MHC class II (especially sentences regarding minor histocompatibility complex), 4.4.1. FlowPRA, and 4.4.2. Single antigen beads assay (SAB).

Specific issues

Point 2. Introduction, page 2, line 7: “evidence to suggest that g-DSA might play an important role in the diagnosis, prognosis,…” this statement appears somewhat exaggerated, since the method has been published exclusively by the review authors themself. In addition, the fact that the pharmaceutical company marketing this method is represented among the authors, is raising doubts on the objectivity of this statement.

Response 2.:

Thank you for your important indication. We agree that this is not major procedure, reported by many institutions. Thus, we simply deleted this sentence “There is clear evidence to suggest that g-DSA might play an important role in the diagnosis, prognosis, and estimation of therapeutic responses.”.

Point 3. Page 2, line 31 “Currently, the main HLA identification tool is the DNA typing technique (rSSOP method) based on the Luminex system (14)”.

The basis of this statment is not clear: The reference is aged 10 years. It is not clear whether the statement refers to high- or low-resolution typing. NGS in case of HR typing, and rSSOP as well as SSP low resolution should be mentioned, as well as the practicability and suitability of the respective methods with respect to the available time in the clinical setting.

Response 3.:

We appreciate your helpful comment. We also agree this sentence is somewhat confusing. To avoid confusion, we would like to simply mention that these methods are available as a HLA identification tool. Thus, we deleted the latter half of sentences in 2.1. We terminated this paragraph at the following sentences.

“At the present day, it is known facts that HLA-D is not a specific antigen but instead is composed of HLA-DR, DQ, and DP(12). Several methods have been available for HLA typing, such as sequence specific primers (SSP), reverse sequence specific oligonucleotide probes (rSSOP) sequencing based typing (SBT), and next-generation sequencing (NGS)”

Point 4. page 3, lines 12, 23

“On the otherhand, HLA-E, HLA-F, HLA-G, NKG2D ligand receptor and FcRn are classified as nonclassical MHC class I (27-29).”

NKG2D does not belong to the non-classical MHC class I. Instead, it is a receptor for the latter.

Response 4.:

Thank you for your correction.We deleted “NKGD ligand receptor” and changed it to “MHC class I polypeptide-related sequence A”.

We wish to thank the reviewer again for your valuable comments.

Reviewer 2 Report

In this review, Nakamura et al. provided a comprehensive review on HLA testing in transplant.  Currently, the field of transplant diagnostics is hyper-focused on antibody detection and methodology related to antibody detection; however, the authors discuss, often forgotten, the function of MHC and how it may drive an immune response.  In addition, the authors highlight recent work by their group and others associated with ICFA- which is an interesting and potentially important diagnostic tool worth discussing.

A very minor critic is section 4.4.1 FlowPRA

The authors should briefly discuss the logic and panel composition used to determine frequency in the PRA assay.

Author Response

Dear Reviewer 2

Thank you for the valuable and helpful comments, which have helped us to improve the manuscript.

In this review, Nakamura et al. provided a comprehensive review on HLA testing in transplant.  Currently, the field of transplant diagnostics is hyper-focused on antibody detection and methodology related to antibody detection; however, the authors discuss, often forgotten, the function of MHC and how it may drive an immune response.  In addition, the authors highlight recent work by their group and others associated with ICFA- which is an interesting and potentially important diagnostic tool worth discussing.

A very minor critic is section 4.4.1 FlowPRA

The authors should briefly discuss the logic and panel composition used to determine frequency in the PRA assay.

Response:

We appreciate your helpful comment. We have corrected the section (4.4.1 FlowPRA) in order to convey the logic and panel composition as followes.

“4.4.1. FlowPRA screening test is performed to detect anti-HLA antibodies in the serum. FlowPRA uses latex beads coated with mixed class I or class II HLA antigens purified from panel cells. About 30 types of panel cells are selected for class I and class II respectively to cover typical types of HLA antigens (One Lambda Inc, Canoga Park, CA). First, a reaction is caused between anti-HLA antibodies and these beads. Next, FITC-conjugated anti-human IgG is added. Then, mean fluorescence intensity (MFI) is measured by flow cytometry and the shift from the negative control is calculated. The specificity of anti-HLA antibodies cannot be identified by FlowPRA, because each bead contains multiple HLA antigens.”

We wish to thank the reviewer again for your valuable comments.

Round 2

Reviewer 1 Report

It seems the authors were able to address all of the issues appropriately!